# High Flow in the Storm. Early Administration of High-Flow Nasal Cannula in Patients with Severe Acute Hypoxic Respiratory Failure Due to Clinically Suspected COVID-19

Sara Jimeno [1,2], Máximo Gómez [2,3], Paula Sol Ventura [2,4,5], Ángeles Calle [1,2], Elena Núñez [2,6], José María Castellano [2,7,8] and Alejandro López-Escobar [9,*]

1    Pediatrics Department, Hospital Universitario HM Puerta del Sur, 28938 Madrid, Spain; sarajimeno@hotmail.com (S.J.); angelescalle@hotmail.com (Á.C.)
2    Fundación de Investigación HM Hospitales, 28015 Madrid, Spain; mgf3566@gmail.com (M.G.); paulasolventura@hotmail.com (P.S.V.); elenanncc@gmail.com (E.N.); jmcastellano@fundacionhm.com (J.M.C.)
3    Neumology Department, Hospital Universitario HM Puerta del Sur, 28938 Madrid, Spain
4    Pediatrics Department, Hospital Universitario HM Nens, 08009 Barcelona, Spain
5    Fundacio Institut d'Investigacio en Ciències de la Salut Germans Trias i Pujol (IGTP), 08916 Badalona, Spain
6    Internal Medicine Department, Hospital Universitario HM Puerta del Sur, 28938 Madrid, Spain
7    Cardiology Department, Hospital Universitario HM Montepríncipe, 28660 Madrid, Spain
8    Centro Nacional de Investigaciones Cardiovasculares, Instituto de Salud Carlos III, 28029 Madrid, Spain
9    Pediatrics Department, Hospital Vithas Madrid La Milagrosa, Unidad de Investigación Clínica, Fundación Vithas, 28010 Madrid, Spain
*    Correspondence: LopezEA@vithas.es

**Abstract:** Background: The worldwide COVID-19 pandemic has created a shortage of ICU beds and ventilators. The objective was to assess whether administration of high-flow nasal cannula (HFNC) in patients with acute hypoxic respiratory failure due to COVID-19 averted mechanical ventilation (MV). Methods: Prospective observational study performed at Hospital Universitario HM Puerta del Sur (Madrid). The protocol included early administration of HFNC in clinically suspected COVID-19 patients with progressive desaturation. Results: Twenty patients were started on respiratory support with HFNC. Hospital admission took place after a median of 7 days since symptom onset and clinical deterioration was apparent at 9 days after symptom onset. Anti-inflammatory treatment with methylprednisolone and tocilizumab was initiated at 9 days (6.5–12), followed by HFNC at 9.5 days (7–12). HFNC was maintained for an average of 4.5 days (2.8–6.3), was successful in eighteen patients (90%), as defined by not needing invasive MV, and failed in two cases (10%) resulting in death. Since HFNC was implemented, there has been a decrease in the number of patients admitted to the ICU and treated with MV for acute hypoxic respiratory failure. Conclusions: HFNC administration may represent a viable therapeutic option for patients in the early stages of severe respiratory failure due to clinically suspected COVID-19.

**Keywords:** oxygen; nasal cannula; respiratory insufficiency; hypoxia; COVID-19

## 1. Introduction

In December 2019, a novel coronavirus, named severe acute respiratory syndrome coronavirus 2 (SARS-CoV-2), emerged in Wuhan, China [1]. Since then, it has caused a worldwide pandemic, with vast health and economic repercussions. The disease, termed COVID-19 by the WHO, ranges from a mild self-limiting form of the disease to severe progressive pneumonia, acute respiratory distress syndrome (ARDS), sepsis and septic shock, multiple organ failure, and death [2–5].

One of the key clinical respiratory manifestations of COVID-19 is hypoxic respiratory failure, a severe arterial hypoxemia that is refractory to supplemental oxygen. Recent reports estimate that approximately 15–30% of patients will require additional oxygen

therapy [1–4] and 5% will require admission to the ICU and mechanical ventilation, with an elevated mortality rate [6]. While it is still unclear why some patients develop ARDS, various risk factors have been described, including male sex, older age, and the coexistence of comorbidities such as diabetes, cardiovascular disease, chronic obstructive pulmonary disease, and hypertension [6].

In January 2020, the first case of COVID-19 was reported in Spain, and the first death due to COVID-19 was reported in mid-February 2020. The growth of confirmed cases has been exponential since then, making Spain one of the most affected countries worldwide. Madrid has been the most affected region, imposing high pressure on the health system. Considering the current ignorance of a proven pharmacological treatment, symptomatic support is the main therapeutic strategy, especially respiratory support.

With the objective of investigating effective strategies for improving the prognosis of patients with progressive hypoxemic respiratory failure due to COVID-19, we undertook a prospective observational study to assess whether the early administration of high-flow nasal cannula (HFNC) averted mechanical ventilation and hence admission to the ICU.

## 2. Materials and Methods

This was a prospective observational study performed at Hospital Universitario HM Puerta del Sur, Madrid, Spain, which included 20 patients from 16 March to 6 April 2020. SARS-CoV-2 infection was confirmed at admission by real-time reverse transcription polymerase chain reaction (RT-PCR) assay. During the study period, due to the dramatic pandemic situation with a multitude of admitted patients and a shortage of PCR tests, there were changes in the diagnostic protocol by the Spanish Ministry of Health. For several weeks, the diagnosis of COVID-19 was based solely on clinical characteristics and chest X-ray imaging.

All patients were treated according to current national standard protocols, which at the time of the study consisted of administration of lopinavir/ritonavir plus hydroxychloroquine substituting lopinavir/ritonavir for azithromycin in patients presenting a prolonged QT interval. In cases of respiratory deterioration in patients with increased respiratory distress and increasing oxygen needs, tocilizumab and methylprednisolone were administered. Regarding support measures, supplemental oxygen therapy was administered based on oxygen saturation levels, scaling from nasal prongs to an oxygen mask with fraction of inspired oxygen (FiO2) up to 50% and eventually a mask with a reservoir. In cases of additional deterioration of oxygen saturation, patients were admitted to the ICU where intubation and mechanical ventilation was started. The national standard protocol included early administration of HFNC in patients with progressive desaturation despite oxygen administration with a reservoir 100% FiO2, with the objective of preventing further respiratory failure and hence admission to the ICU. The study protocol was approved by the HM Hospitals Local Ethics Committee (approval number 20.03.1573-GHM), and all participants gave verbal informed consent in the presence of witnesses and the main investigator, this fact being reflected in the medical history.

Patients included in the study had to meet the following inclusion criteria: (1) age between 18 and 79 years, (2) positive detection of SARS-CoV-2 by pharyngeal/nasal swab PCR or clinical or radiological suspicion of COVID-19, (3) peripheral oxygen saturation (SpO$_2$) <90% receiving oxygen therapy through a mask with a reservoir and oxygen flow at 15 lpm, and (4) suitable to verbally accept informed consent.

Exclusion criteria included the need of ICU and immediate mechanical ventilation due to significant deterioration of respiratory function with tachypnea over 35 rpm, intense desaturation and/or stuporous state of consciousness, participation in another clinical trial, or the presence of any physical or mental condition which, at the discretion of the investigator, contraindicated enrollment. Patients with COVID-19 develop hypoxemic respiratory failure, not so much ventilatory. When we performed an arterial blood gas in some patients, we confirmed hypoxemia without observing hypercapnia or respiratory acidosis. For this reason, we did not consider including blood gas as a criterion for

inclusion or exclusion of patients in our study. Patients received HFNC with AIRVO™ 2 (Fisher & Paykel Healthcare, Auckland, New Zealand). Due to the shortage of devices, MR850 respiratory humidifiers (Fisher & Paykel Healthcare, Auckland, New Zealand) were adapted with flowmeters to supply up to 60 lpm of humidified and hot oxygen through heated breathing tube AirSpiral™ (Fisher & Paykel Healthcare, Auckland, New Zealand) and Optiflow™ interfaces (Fisher & Paykel Healthcare, Auckland, New Zealand). HFNC treatment was started by setting the temperature at 37 °C and regulating an initial flow of 30 L/min, which could be increased up to 60 depending on respiratory distress and patient tolerance. The $FiO_2$ was set to maintain $SpO_2$ levels above 92%. In this very early phase of the pandemic, there was no established protocol for treating COVID-19. HFNO treatment was continuous, with close monitoring of vital signs and respiratory patterns. Weaning from HFNC was attempted when the patient began to improve clinically (oxygenation and subjective sensations of the patients) and radiologically (reduction in pulmonary infiltrates). Clinical improvement was defined as $SpO_2 > 92\%$ mask with a reservoir without dyspnea, allowing the passage of oxygen therapy through a reservoir at 15 lpm. HFNC failure was defined as the need for mechanical ventilation due to deterioration of respiratory function or death of the patient.

To minimize the potential risk of virus transmission, all hospital healthcare workers treated clinically suspected COVID-19 patients with personal protective equipment (gown, hat, mask, protective screen, or glasses, gloves). All patients with clinically suspected COVID-19, and especially those treated with HFNC, were instructed to wear a surgical mask during the hospital stay. Additionally, the hospital implemented negative pressure measures from the beginning of the pandemic.

The following information was collected for all patients: age, gender, date of onset of symptoms and hospital admission, the coexistence of comorbidities (including risk factors before mentioned), laboratory test results (including full blood count, C-reactive protein, D-dimer), vital signs including breathing frequency, radiological progression, duration of HFNC, duration of $O_2$ therapy, length of stay, and outcome (hospital discharge, ICU admission, or death).

Continuous variables were reported as median value and interquartile range (IQR) or range minimum–maximum when appropriate. The differences between groups were analyzed by Mann–Whitney U test. Categorical variables were reported as number and percentage and analyzed using the chi-squared test and Fisher's exact test. A *p* value < 0.05 was considered significant.

## 3. Results

*Subsection*

- During the study period, from 16 March to 6 April 2020, 358 patients were admitted at Hospital Universitario HM Puerta del Sur, due to clinically suspected COVID-19 infection (Figure 1). A total of 27 (7.5%) patients experienced severe acute respiratory failure, of which 7 (1.9%) were admitted to the ICU requiring invasive mechanical ventilation immediately, while 20 (5.5%) met inclusion criteria and were enrolled, and hence were started on respiratory support with HFNC. A total of 11 (55%) patients had positive detection of SARS-CoV-2 by pharyngeal/nasal swab PCR. The clinical characteristics of all 20 patients receiving HFNC at hospital admission are summarized in Table 1. Hospital admission took place after a median of 7 days (5.3–9.8) from symptoms onset. At the time of admission, all patients received the standard approved treatment for COVID-19 as previously described. Clinical and radiological worsening occurred at 9 days (7–10.8) and 9 days (7–10), respectively, after the symptom onset, which initiated anti-inflammatory treatment with methylprednisolone and tocilizumab at 9 days (6.5–12) and 9 days (7–10.3), respectively, as well as HFNC at 9.5 days (7–12). HFNC was maintained for an average of 4.5 days (2.8–6.3) (Figure 2). Of the total population receiving HFNC, two (10%) patients died, and eighteen (90%) patients did not require mechanical ventilation as rescue therapy. Of

the HFNC successful group, fifteen patients were discharged, two improved but were still admitted without HFNC and decreasing oxygen administration and one of them continued HFNC after 19 days. Only one patient failed to wean from HFNC, improving after a new period with HFNC. In the two patients in whom HFNC failed, the treatment was well tolerated and was not discontinued. Death was a consequence of refractory hypoxemia. At the time of HFNC failure, none of the patients was a candidate for mechanical ventilation.

- At the time of hospital admission, no significant clinical differences were observed between patients in whom HFNC was successful in preventing respiratory deterioration and those in whom it failed; there were no treatment differences or HFNC characteristics between both groups (Table 1).
- Laboratory findings are summarized in Table 2. Subjects who died showed a significantly lower neutrophil count on the first day of admission 2445/mm$^3$ (2090–2800/mm$^3$) vs 7410/mm$^3$ (4757–9510/mm$^3$) ($p$ = 0.032), and subsequently, a significantly greater increase in the peak neutrophil lymphocyte ratio (NLR) 47.9 (33.3–62.6) vs. 14.8 (11–24.3) ($p$ = 0.044). Furthermore, the peak NLR was observed after a few days of worsening, which occurred significantly faster in patients who subsequently died compared with the rest: 16.1 (4.3–27.8) vs. 1.6 (4.3–27.8) ($p$ = 0.025) (Table 2).
- Since the implementation of HFNC, the number of patients requiring mechanical ventilation due to acute respiratory failure decreased significantly (Figure 3).

**Table 1.** Clinical characteristics. Intervention and HFNC success group: median value (interquartile range (IQR)); HFNC failure group: median (range). FiO$_2$ (%): fraction of inspired oxygen concentration set. PCR test by pharyngeal/nasal swab. Intervention group: 20 patients who met the inclusion criteria and were included in the respiratory support group with HFNC.

| | Intervention 20/20 | HFNC Success n = 18 | HFNC Failure n = 2 | *p* |
|---|---|---|---|---|
| Failure of treatment and mortality (%) | 20 (100) | 0 (0) | 2 (100) | 0.005 |
| Age (years) | 63.5 (47–71) | 62 (45–69) | 76 (72–80) | 0.063 |
| Male (%) | 10 (50) | 9 (50) | 1 (50) | 1 |
| Comorbidity (%) | 11 (55) | 9 (50) | 2 (100) | 0.479 |
| Positive PCR test | 11 (55) | 9 (50) | 2 (100) | 0.479 |
| Signs and symptoms (hospital admission) | | | | |
| Fever (%) | 17 (85) | 15 (83) | 2 (100) | 1 |
| Cough (%) | 17 (85) | 16 (89) | 1 (50) | 0.284 |
| Dyspnea (%) | 17 (85) | 17 (94) | 0 (0) | 0.016 |
| Odynophagia (%) | 2 (10) | 2 (11) | 0 (0) | 1 |
| Myalgia (%) | 5 (25) | 4 (22) | 1 (50) | 0.447 |
| SpO$_2$ (%) | 92 (85–92) | 91 (64–95) | 95 (93–98) | 0.095 |
| Clinical Evolution | | | | |
| Day of symptoms until hospital admission | 7 (5.3–9.8) | 7 (6–10.3) | 4.5 (4–5) | 0.095 |
| Day of symptoms until clinical worsening | 9 (7–10.8) | 9 (7–11.3) | 6.5 (6–7) | 0.168 |
| Day of symptoms until radiological worsening | 9 (7–11) | 9 (7–11) | 6.5 (6–7) | 0.126 |
| Day of symptoms until methylprednisolone | 9 (6.5–12) | 10 (7–13) | 6.5 (6–7) | 0.176 |
| Day of symptoms until tocilizumab | 9 (7–10.3) | 9 (7.3–10.8) | 6.5 (6–7) | 0.118 |
| HFNC setting | | | | |
| Day of symptoms until start HFNC | 9.5 (7–12) | 9.5 (7–12.5) | 9 (8–10) | 0.853 |
| Day of symptoms until finish HFNC | 14 (11.8–18) | 14.5 (12–18) | 12 (10–14) (Death) | 0.392 |
| Duration of HFNC, days | 4.5 (2.8–6.3) | 5 (3–6.7) | 3 (2–4) (Death) | 0.327 |

**Table 2.** Intervention and HFNC success group: median value (interquartile range (IQR)); HFNC failure group: median (range). FiO$_2$ (%): fraction of inspired oxygen concentration set. NLR: neutrophil-lymphocyte rate. * Corresponding value at NRL maximum.

| | Intervention 20/20 | HFNC Success n = 18 | HFNC Failure n = 2 | p |
|---|---|---|---|---|
| Failure of treatment and mortality (%) | 20 | 0 (%) | 2 (100%) | 0.005 |
| Laboratory tests | | | | |
| Neutrophils (count/mm$^3$) admission day | 6205 (4165–9287) | 7410 (4757–9510) | 2445 (2090–2800) | 0.032 |
| Lymphocyte (count/mm$^3$) admission day | 770 (530–1150) | 770 (535–1210) | 660 (300–1200) | 0.573 |
| NLR admission day | 11 (3.6–12.1) | 11.1 (4.8–12.3) | 4.9 (2.8–7) | 0.184 |
| Neutrophils (count/mm$^3$) NLR maximum | 10,835 (9135–14,152) | 10,835 (8800–13,877) | 13,139 (11,120–16,259) | 0.853 |
| Lymphocyte (count/mm$^3$) NLR maximum | 590 (452–827) | 635 (515–847) | 325 (160–490) | 0.095 |
| NLR maximum | 15.2 (11.3–27.7) | 14.8 (11–24.3) | 47.9 (33.32–62.6) | 0.044 |
| D of S until INL maximum | 15 (9.5–18.8) | 15.5 (10.5–19.3) | 9 (7–11) | 0.211 |
| NLR termination | 4 (1.9–12.6) | 4 (1.9–12.6) | (Death) | np |
| NLR increasing speed/day | 1.7 (0.5–3) | 1.6 (0.5–2.7) | 16.1 (4.3–27.8) | 0.025 |
| C-reactive protein (mg/L) * | 14.6 (6–86) | 11.1 (6–73) | 56.8 (14.6–99) | 0.529 |
| D-dimer (ng/ml) * | 1694 (1018–15,854) | 1094 (958–14,460) | 9859.5 (1385–18,334) | 0.491 |

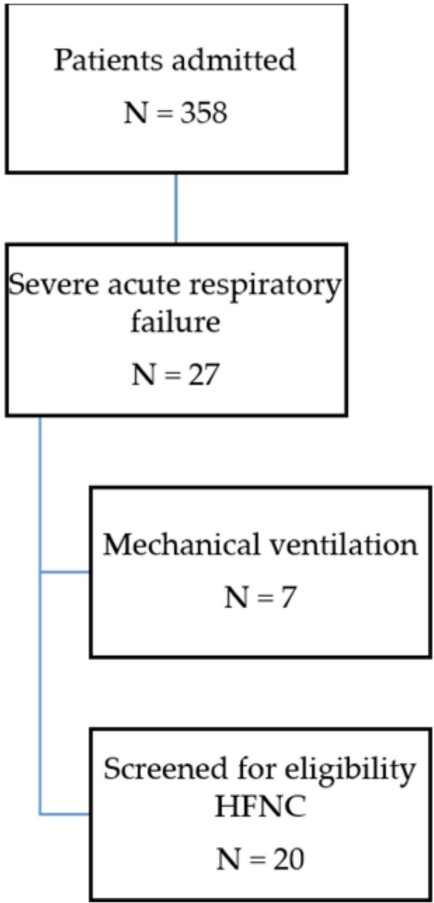

**Figure 1.** Flow of patient screening and enrollment.

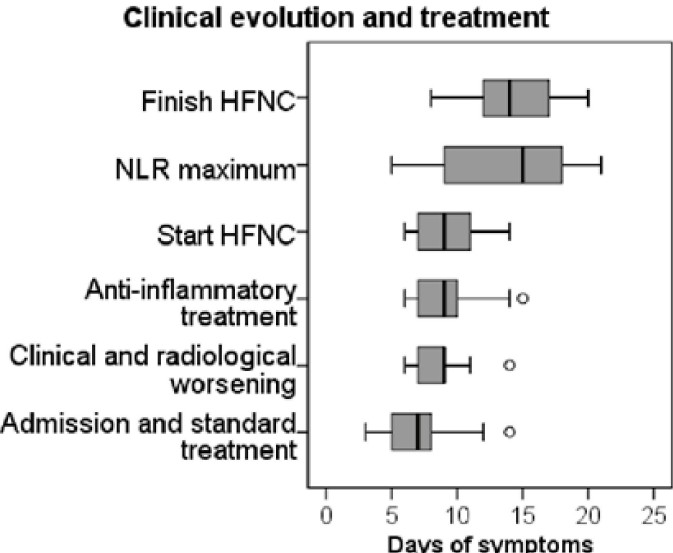

**Figure 2.** Clinical evolution of the patients (days with respect to the onset of symptoms) according to admission, treatments, and clinical and laboratory worsening.

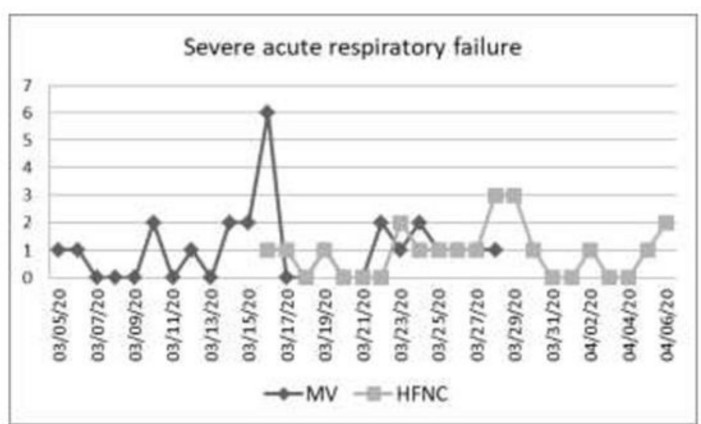

**Figure 3.** Patients admitted to hospital treated with mechanical ventilation (MV) and HFNC during period of study. Since the implementation of HFNC, the number of patients requiring mechanical ventilation due to acute respiratory failure decreased significantly.

## 4. Discussion

At the beginning of the pandemic, on 16 March 2020, when the present study began, there was a very significant limitation of ICU beds and ventilators in Spain, and this was one of the reasons that led the research team to develop strategies that could improve the clinical evolution of a greater number of patients. Considering the patients in whom this therapy was used, many of them were not eligible to receive invasive therapy due to the ethical principle of justice, much less after already having received a non-invasive therapy; but, more invasive therapy would not have prevented a refractory hypoxemic failure that would have benefited from prolonged mechanical ventilation in the ICU.

In the first months of the global pandemic, there had not been any reports on the usefulness of HFNC as a support therapy for respiratory failure caused by COVID-19 infection, and there was much controversy about the use of HFNC therapy due to the generation of aerosols. At that time, invasive ventilation tended to be applied very early in patients with COVID-19 who worsened. Treatment of respiratory failure due to other causes with HFNC has gained popularity in recent years. The results of a systematic review and meta-analysis of nine RCTs showed that HFNC reduces the need for mechanical

ventilation compared with conventional oxygen therapy (RR 0.85, 95% CI 0.74 to 0.99) in a population of patients with acute hypoxemic respiratory failure, although the use of this technique did not affect the length of stay in the ICU or the risk of death [7,8]. The way in which these results in patients with ARDS of undifferentiated etiology are applicable to COVID-19 was unknown in the first months of the pandemic. A posteriori, in May 2020, The Surviving Sepsis Campaign: Guidelines on the Management of Critically Ill Adults with Coronavirus Disease 2019 (COVID-19) suggests the use of HFNC over conventional oxygen therapy for adults with COVID-19 and acute hypoxemic respiratory failure despite conventional oxygen therapy [9]. In recent months, various case series with COVID-19 that applied this treatment have been published with satisfactory results [4,10–12].

The biophysics of droplets and gas cloud formation and its implications for patients infected with the SARS-CoV-2 virus have been reported as a potential concern when using HFNC in COVID-19 patients due to the possibility that viruses may be aerosolized, resulting in higher exposure to healthcare workers during endotracheal intubation [13–15]. Very low-certainty evidence showed uncertain findings with regard to droplet dispersion and aerosol generation with HFNC [16]. In fact, a recent study shows that HFNC, compared with oxygen therapy with a mask, does not increase dispersion or microbiological contamination in the environment; bio-aerosol dispersion via HFNC involves a risk similar to that of standard oxygen masks [17]. However, in order to minimize this potential risk, we started several safety measures, including implementing negative pressure measures in the hospital since the beginning of the pandemic, as well as the use of surgical masks in all patients.

Clinical data from past MERS-CoV and SARS-CoV infections has shown that β-coronaviruses infections can cause fatal lower respiratory tract infections as well as various extrapulmonary manifestations [18,19]. After all these months of worldwide pandemic, the pathophysiology and current understanding of COVID-19 disease allows differentiating between three distinct phases of disease progression [20]. The initial phase includes SARS-CoV-2 replication, primarily in the respiratory system. Most patients present mild and non-specific symptoms such as fever and dry cough. This phase is followed by a well-established pulmonary disease presenting with bilateral infiltrates or ground glass opacities in the chest X-ray and computerized tomography. Blood count alterations become apparent at this stage, mainly in the form of lymphopenia, and may signal the start of a severe inflammatory response that has been called cytokine storm syndrome, resulting in interstitial pneumonitis with lymphopenia and increased presence of neutrophils in alveolar tissue. The final phase includes systemic hyper-inflammation with severe multiorgan damage. The cellular and molecular pathophysiology of the disease has potential profound implications in implementing treatment strategies targeted toward each specific disease stage [21–23].

In our study, 7.5% of patients admitted for clinical suspicion of COVID-19 experienced severe acute respiratory failure. This percentage of severe acute respiratory failure (8%) is similar to that observed in another publication of our research group, a study that was carried out on a larger sample, 2254 patients [24].

Patients included in the current study showed a similar clinical course as previously described: admission took place on day 7 after the onset of symptoms; reason for admission was shortness of breath, as well as presence of other mild and non-specific symptoms (Table 1) [2]. Although we did not carry out specific inflammatory mediators' measurements, clinical worsening at 9 days after symptom onset requiring higher oxygen demands correlated with signs of radiological worsening in the form of infiltrates in the chest X-ray, which may be signaling progressive inflammatory parenchymal infiltration in the lung. The presence of infiltrates currently establishes the point at which anti-inflammatory treatment should be started (Figure 2). At this point, subjects develop ARDS, which is mainly hypoxemic, and therefore the need of ventilator support until recovery or death. HFNC was initiated at 9 days and maintained for 4.5 days, providing support while anti-inflammatory treatment took effect and avoiding ventilator-induced lung injury due to mechanical ventilation [25]. Administration of oxygen therapy through a device that heats

and humidifies the inspired gas could help decrease damage since it preserves the nasal mucosa and improves ciliary movement and clearance of secretions, prevents bronchial hyper-responsiveness associated with inhalation of cold and dry gas, and controls the energy expenditure required to condition the inspired air [26].

HFNC failed in two cases (10%). The evolution and characteristics of these patients made them not candidates for admission to the ICU. We hypothesize that these two subjects developed a more severe cytokine storm syndrome that may have precipitated this outcome. Interestingly, the two patients that died during the study presented lower neutrophil count on the first day of admission and significantly higher peak NLR compared with the rest of the population. Furthermore, the rate of increase in NLR was highest in both as well, supporting the idea of a greater hyper-inflammatory response, despite receiving anti-inflammatory treatment with both corticosteroids and tocilizumab (Table 2). Our results are in accordance with several studies published on the clinical characteristics of patients with SARS-CoV-2 pneumonia, and they show that severely ill patients tend to have a higher proportion of NLR [23,27,28].

Several published works describing retrospective series have demonstrated the beneficial effect of HFNC on COVID-19. Yang et al. published findings on a sample of 33 patients treated with HFNC, with a failure rate of 48% [10], and Wang et al. reported a sample of 4 patients with a failure rate of 41% [4]. More recently, Xia et al. published a report on a sample of 43 patients, and the HFNC failure rate was 46.5% [11]. According to our results, the use of HFNC avoided mechanical ventilation in 85% of cases, which may be related to prompt administration of HFNC in the earlier stages of clinical and radiological worsening. HFNC may maintain respiratory function, and combined with anti-inflammatory treatment, prevent invasive ventilation in a significant proportion of patients with ARDS.

This study has several limitations. First, the small number of patients is an inherent limitation, and we have not compared outcomes with non-invasive ventilation or invasive ventilation. With such a low number of patients, we could not pretend to establish criteria for use or recommend parameters or establish a treatment guideline or protocol when nothing was known about HFNC and COVID, but rather can only convey a special situation at a very specific time, conditioned by external factors never seen before. Second, SARS-Cov-2 infection was confirmed by PCR in only 55% of included patients; the rest of the patients presented clinical evolution, laboratory findings, and radiological pattern compatible with COVID-19. Furthermore, the clinical and radiological course was similar in all patients, with its characteristic temporal pattern. Hence the findings should be interpreted with caution. Undoubtedly, this may serve as a reference when conducting larger randomized clinical trials.

Finally, it is possible that it would have been useful to perform blood gas analysis before starting therapy and to assess its gasometric repercussions, but clinical experience showed us that patients with severe hypoxemia did not usually present hypercapnia in blood gases. This led us to not carry out a serial monitoring of blood gases that would have been unnecessary and painful for patients throughout their evolution.

## 5. Conclusions

Our study supports the early use of HFNC in patients with severe respiratory failure due to clinically suspected COVID-19, which may represent a viable therapeutic measure, combined with anti-inflammatory treatment, which could avoid mechanical ventilation and the exposure of healthcare personnel to the risk of contagion.

## 6. Patents

No patents resulting from the work reported in this manuscript.

**Author Contributions:** Literature search: S.J., Á.C. and A.L.-E. Data collection: S.J., Á.C. and A.L.-E. Study design: P.S.V., J.M.C. and A.L.-E. Analysis of data: P.S.V., J.M.C. and A.L.-E. Manuscript preparation and writing: S.J., Á.C. and A.L.-E. Review of manuscript and approval of the version to

be published: S.J., M.G., P.S.V., Á.C., E.N., J.M.C. and A.L.-E. All authors have read and agreed to the published version of the manuscript.

**Funding:** This research received no external funding.

**Institutional Review Board Statement:** The study protocol was approved by the HM Hospitals Local Ethics Committee (approval number 20.03.1573-GHM).

**Informed Consent Statement:** Verbal informed consent in the presence of witnesses and the main investigator was obtained from all subjects involved in the study, this fact being reflected in the medical history.

**Conflicts of Interest:** The authors declare no conflict of interest.

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
