# Peer review of "High Flow in the Storm. Early Administration of High-Flow Nasal Cannula in Patients with Severe Acute Hypoxic Respiratory Failure Due to Clinically Suspected COVID-19"

_2673-8430, doi:10.3390/biomed1020012_

Round 1

Reviewer 1 Report

Thank you for giving me the opportunity to review the Manuscript by Sara Jimeno and coauthors on high flow nasal cannula (HFNC) in patients with covid-19.

The authors investigate an important subject since HFNC is well tolerated and offers a very effective way of oxygen delivery to conscious and spontaneous breathing hypoxic patients. The authors repot that 90% of included patients could be managed with HFNC only, while in 10% HFNC failed and the patients died. The authors conclude that HFNC administration may represent a viable therapeutic option for patients in the early stages of severe respiratory failure due to COVID-19.

I have the following major remarks

Only 27/358 (8%) of patients with clinical suspected covid-19 admitted to the hospital were put on HFNC or IMV, which sounds low compared to other published registries. Please provide additional data on patients not included since exclusion criteria should allow for high recruitment of patients.

The endpoint of the present study was respiratory recovery. The methods section however does not state how the clinical improvement (defined as SpO2 > 92% mask with a reservoir without dyspnea) was tested. Was there an algorithm? How many patients failed weaning and had to be put back on HFNC?

The cause of death should be given for the two patients failing HFNC. It might be important to investigate if therapy was withdrawn, patients develop refractory hypoxia or hypercapnia or if they did not tolerate the HFNC and had to be sedated and intubated.

Since NHFC was investigated, data on the actual therapy should be given. Average flow and oxygen saturation, respiratory rate during high flow, time /day spend on therapy, etc.. Since hypercapnia is not effectively controlled by HFNC, apCO2 / pH of patients should be investigated. Were the two patients who failed HFNC hypercapnic, perhaps even at the start since blood gases were not included in the inclusion/exclusion criteria?

Covid 19 ARDS is a disease which typically lasts more than 4 days, especially in the subgroup of patients with high oxygen demand as included in the present study. Knowing that covid was not proven by PCR in half of the patients, do the authors have arguments that these patients actually had covid?

The authors state in the abstract that “since HFNC was implemented, there has been a decrease in the number of patients admitted to the ICU and treated with MV for acute hypoxic respiratory failure.” This statement should be sounded with data or removed.

Since 55% of patients were diagnosed clinically and no SARS-PCR has been performed, this represents a significant bias in the study. I strongly suggest replacing “covid-19” with “clinically suspected covid-19” through the manuscript and the title. Also, number of positive PCR results have to be included in table 1 for each group.

The authors test significance in 2 groups with one group having only 2 patients and do this for roughly 30 items. I suggest either giving sound statistical reasoning for this unconventional testing or better to remove the significance calculations from the manuscript (or at least to limit testing to viewer items and include the Bonferroni correction).

Author Response

  1. Only 27/358 (8%) of patients with clinical suspected covid-19 admitted to the hospital were put on HFNC or IMV, which sounds low compared to other published registries. Please provide additional data on patients not included since exclusion criteria should allow for high recruitment of patients.

The present study was carried out very early in the pandemic in Spain, from March 16 to April 6, 2020. At that time, the beneficial role of HFNC in COVID-19 was not so clear. Then, we selected only those patients with progressive desaturation despite the administration of oxygen with a 100% FiO2 reservoir. Exclusion criteria included the need of ICU and immediate mechanical ventilation due to significant deterioration of respiratory function with tachypnea over 35 rpm, intense desaturation and / or stuporous state of consciousness, participating in another clinical trial or the presence of any physical or mental condition which, at the discretion of the investigator, contraindicated enrollment.

However, this percentage of severe acute respiratory failure (8%) is similar to that observed in another publication of our research group, a study that was carried out on a larger sample, 2254 patients.

Velázquez S, Madurga R, Castellano Vázquez JM, et al. Hemogram rate as

prognostic markers of Care Unit Admission in COVID-19. MC Emergency Medicine. 2021;21(1),1-9. DOI: 10.21203/rs.3.rs-403472/v1.

  1. The endpoint of the present study was respiratory recovery. The methods section however does not state how the clinical improvement (defined as SpO2 > 92% mask with a reservoir without dyspnea) was tested. Was there an algorithm? How many patients failed weaning and had to be put back on HFNC?

In this very early phase of the pandemic, there was no established protocol for treating COVID-19. Weaning from HFNC began when the patient began to improve clinically and radiologically. Clinical improvement was considered in those patients who presented SpO2> 92% with a mask with a reservoir without clinical signs of dyspnea. Only one patient failed to wean from HFNC, improving after a new period with HFNC. Two patients did not allow the initiation of a decrease in HFNC with failure of therapy and fatal outcome.

  1. The cause of death should be given for the two patients failing HFNC. It might be important to investigate if therapy was withdrawn, patients develop refractory hypoxia or hypercapnia or if they did not tolerate the HFNC and had to be sedated and intubated.

In two patients in whom HFNC failed, the treatment was well tolerated and was not discontinued. Death was a consequence of refractory hypoxemia. At the time of HFNC failure, none of the patients was a candidate for mechanical ventilation.

  1. Since NHFC was investigated, data on the actual therapy should be given. Average flow and oxygen saturation, respiratory rate during high flow, time /day spend on therapy, etc.. Since hypercapnia is not effectively controlled by HFNC, apCO2 / pH of patients should be investigated. Were the two patients who failed HFNC hypercapnic, perhaps even at the start since blood gases were not included in the inclusion/exclusion criteria?

HFNC treatment was started by setting temperature at 37ºC and regulating an initial flow of 30 L/min, that could be increased up to 60 depending on respiratory distress and patient tolerance. The FiO2 was set to maintain SpO2 levels above 92%.

What was observed in patients with COVID-19 was that they developed hypoxemic respiratory failure, not so much ventilatory, for this reason we did not perform unnecessary arterial blood gases. For this reason, we do not consider including blood gas as a criterion for inclusion or exclusion.

Patients with HFNC did not develop hypercapnia or respiratory acidosis during treatment. The two patients in whom HFNC failed died as a consequence of refractory hypoxemia.

  1. Covid 19 ARDS is a disease which typically lasts more than 4 days, especially in the subgroup of patients with high oxygen demand as included in the present study. Knowing that covid was not proven by PCR in half of the patients, do the authors have arguments that these patients actually had covid?

We are aware of this important limitation of the study. Due to the dramatic pandemic situation with a multitude of admitted patients and a pressing shortage of PCR tests for all of them, changes were made in the diagnostic protocol established by the Spanish Ministry of Health. For several weeks, the diagnosis of Covid-19 was based solely on clinical features and findings on chest X-rays. SARS-Cov-2 infection was confirmed by PCR in only 55% of the patients included in our study, the rest of the patients presented clinical and radiology compatible with COVID-19. Furthermore, the clinical and radiological course was similar in all patients.

  1. The authors state in the abstract that “since HFNC was implemented, there has been a decrease in the number of patients admitted to the ICU and treated with MV for acute hypoxic respiratory failure.” This statement should be sounded with data or removed.

We greatly appreciate your recommendation. However, in Figure 3 it can be seen that after the implementation of HFNC there is a decrease in the number of patients who required mechanical ventilation due to acute respiratory failure.

  1. Since 55% of patients were diagnosed clinically and no SARS-PCR has been performed, this represents a significant bias in the study. I strongly suggest replacing “covid-19” with “clinically suspected covid-19” through the manuscript and the title. Also, number of positive PCR results have to be included in table 1 for each group.

We greatly appreciate your recommendation.

  1. The authors test significance in 2 groups with one group having only 2 patients and do this for roughly 30 items. I suggest either giving sound statistical reasoning for this unconventional testing or better to remove the significance calculations from the manuscript (or at least to limit testing to viewer items and include the Bonferroni correction).

This factor was taken into account when performing the statistical analysis, therefore there were no statistically significant differences that probably would occur in a larger group of patients. In our study we have tried to describe a sample of patients who did well with the treatment and we tried to find any difference with respect to those who died. Subsequently, we carried out another study in which we expanded the sample to include a greater number of patients.

Jimeno S, Ventura S, Castellano JM, García-Adasme SI, Miranda M, Touza P, et al. Prognostic implications of neutrophil-lymphocyte ratio in COVID-19. Eur J Clin Invest. 2020;e13404. DOI: 10.1111/eci.13404.

Reviewer 2 Report

In this paper the authors aim to verify whether the early administration of high flow nasal oxygen averted mechanical ventilation and therefore admission to the ICU. I have read this work with great interest, the topic is topical and of real usefulness. The work for me is acceptable for posting, however I have a few minor suggestions which in my opinion would make the work clearer and more complete:

1. on line 80 the authors declare "in house of respiratory decline, tocilizumab and methylprednisone were administered", please specify better what is meant by respiratory decline, an increase in respiratory rate? a decrease in saturation values?

2. in table 1 I did not understand what is meant by "intervention"

3. Please implement the caption of figure 2 and 3. The caption should serve as a “stand alone” ie be understandable without going to the text.

4. Finally, if available, it would be interesting to compare the blood gas values of patients before and after treatment with HFNC.

Author Response

  1. on line 80 the authors declare "in house of respiratory decline, tocilizumab and methylprednisone were administered", please specify better what is meant by respiratory decline, an increase in respiratory rate? a decrease in saturation values?

In the case of respiratory deterioration in the patient with increased respiratory distress and increasing oxygen needs, tocilizumab and methylprednisolone were administered.

  1. in table 1 I did not understand what is meant by "intervention"

We considered the "Intervention group" as the 20 patients who met the inclusion criteria and were included in the respiratory support group with HFNC.

  1. Please implement the caption of figure 2 and 3. The caption should serve as a “stand alone” ie be understandable without going to the text.

We greatly appreciate your recommendation.

Figure 2: Clinical evolution of the patients (days with respect to the onset of symptoms) according to admission, treatments, clinical and laboratory worsening.

Figure 3: Since the implementation of HFNC, the number of patients requiring mechanical ventilation due to acute respiratory failure decreased significantly.

  1. Finally, if available, it would be interesting to compare the blood gas values of patients before and after treatment with HFNC.

We greatly appreciate your recommendation but unfortunately we are currently unable to have these values. However, patients with COVID-19 develop hypoxemic respiratory failure, not so much ventilatory, so that throughout the evolution we do not perform unnecessary arterial gases.

Round 2

Reviewer 1 Report

Thank you for giving me the possibility to review revision1 of manuscript. Although the authors provide a reply to all comments made, the actual change in the manuscript remains minimal and therefore most of the major concerns remain. I cannot recommend for publication of a manuscript evaluating high flow in covid-19 when no data on high flow therapy is given and only half of the patents had proven Covid. In research, it is not sufficient to state that all patients presented compatible to Covid, you have to show data. 

Author Response

Dear Sir / Madam,

Thank you very much for the opportunity to improve our manuscript and for your eagerness for us to better expose our research.

We are aware of the important limitations of our work that are highlighted by your insightful and appropriate comments. It is true that we are unable to resolve your major concerns, but this is not an attempt to mask or not communicate, but rather to convey to you and your readers a very specific situation at a very specific moment in time, conditioned by external factors never seen before.

The research shown in this paper would probably have made more sense months ago when it was first written, however in the following months it has been updated according to the available evidence and the current evidence, even with larger series of patients, corroborates our results and our research.

In the first months of the pandemic in Spain, which was the hardest hit country after China and Italy, there was not much data on the disease and there was much controversy about the use of high-flow therapy due to the generation of aerosols and the Sepsis Campaign Guidelines were not very enlightening in this regard. However, we started to use this therapy and the results obtained are as shown. With such a low number of patients we could not pretend to establish criteria for use or recommend parameters or make a treatment guideline when nothing was known about high flow and COVID. That is why it may seem to you a low number of patients treated with high flow, but at that time more conventional mechanical ventilation was used and other factors such as vaccination and the use of other treatments such as corticosteroids earlier, have changed the scenario. However, we do not believe that this influences the conclusions because we did have strict inclusion criteria that are perfectly reflected in the paper.  It is possible that it would have been useful to perform blood gas analysis before starting therapy and assess its gasometric repercussions, but we do not think this could have been done for several reasons: clinical experience told us that the patients had brutal hypoxemia (confirmed with a monitor) and when we performed arterial blood gas analysis in some patients we confirmed hypoxemia without hypercapnia, perhaps due to compensatory polypnea. This led the research team not to perform serial blood gas monitoring because it was considered unnecessary and also painful and invasive for the patient.

Regarding weaning, we did not have an established protocol because there was insufficient evidence for it. The weaning was done as oxygenation improved and allowed lowering the flow from 60 lpm to 45 lpm and finally to 30 lpm. There were no objective criteria except for the improvement of pulmonary infiltrates and oxygenation, and for us the subjective sensations of the patients were very important. We evaluated all the factors together (pulmonary infiltrates, improvement of oxygenation, subjective sensation of comfort by lowering flow) and individually we proceeded to administer oxygen with mask and reservoir without failure in any of the patients in whom we tried to do so (18/20).

Regarding the cause of death of the patients, it was due to multi-organ failure as a consequence of severe respiratory failure. As you may recall, in those first months of the pandemic, specifically in March 2020, in Spain there was a very significant limitation of ICU beds and respirators and this was one of the reasons that led the research team to develop devices that could oxygenate improvement more patients. The patients in whom this therapy was used, many of them were not susceptible to receiving invasive therapy due to the ethical principle of justice, much less after receiving a non-invasive therapy but that did not prevent a refractory hypoxemic failure and that, as you well know, did not would have benefited from prolonged mechanical ventilation in the ICU. Patients in whom it failed did not withdraw, died of respiratory failure, and were not considered candidates for invasive therapy. This extraordinary situation at the national level and I would even say worldwide, as you well know, led to a lack of diagnostic tests for SARS-CoV2 in Spain, which led to the percentage of confirmed patients being the one we present. At the beginning of the pandemic, people stayed home and arrived on many occasions when the window to test positive for the antigen test had passed. All the cases presented had a characteristic radiological pattern of COVID, compatible laboratory findings, and the clinical course with its characteristic temporal pattern was COVID.

Regarding the parameters you request, we do not include it except the duration of the high-flow therapy that is specified in the tables of the article. We did not believe that the rest of the data added value to our research since, as we said previously, we do not intend to establish a treatment guide or protocol. In any case, we could include the vital signs of the subjects and the flow used in each patient if you consider it essential. We could specify this data if you consider it necessary.

I hope with this I have responded to your concerns and trusting in your knowledge and patience, if it is not too much trouble, I would greatly appreciate if you could provide us with the complementary information for the readers of your prestigious magazine.

Thank you for your invaluable help.

Best regards.

The authors.
